# Constructing Deep Neural Networks by Bayesian Network Structure Learning

**Raanan Y. Rohekar**
Intel AI Lab
raanan.yehezkel@intel.com

**Shami Nisimov**
Intel AI Lab
shami.nisimov@intel.com

**Yaniv Gurwicz**
Intel AI Lab
yaniv.gurwicz@intel.com

**Guy Koren**
Intel AI Lab
guy.koren@intel.com

**Gal Novik**
Intel AI Lab
gal.novik@intel.com

## Abstract

We introduce a principled approach for unsupervised structure learning of deep neural networks. We propose a new interpretation for depth and inter-layer connectivity where conditional independencies in the input distribution are encoded hierarchically in the network structure. Thus, the depth of the network is determined inherently. The proposed method casts the problem of neural network structure learning as a problem of Bayesian network structure learning. Then, instead of directly learning the discriminative structure, it learns a generative graph, constructs its stochastic inverse, and then constructs a discriminative graph. We prove that conditional-dependency relations among the latent variables in the generative graph are preserved in the class-conditional discriminative graph. We demonstrate on image classification benchmarks that the deepest layers (convolutional and dense) of common networks can be replaced by significantly smaller learned structures, while maintaining classification accuracy—state-of-the-art on tested benchmarks. Our structure learning algorithm requires a small computational cost and runs efficiently on a standard desktop CPU.

## 1 Introduction

Over the last decade, deep neural networks have proven their effectiveness in solving many challenging problems in various domains such as speech recognition (Graves & Schmidhuber, 2005), computer vision (Krizhevsky et al., 2012; Girshick et al., 2014; Szegedy et al., 2015) and machine translation (Collobert et al., 2011). As compute resources became more available, large scale models having millions of parameters could be trained on massive volumes of data, to achieve state-of-the-art solutions. Building these models requires various design choices such as network topology, cost function, optimization technique, and the configuration of related hyper-parameters.

In this paper, we focus on the design of network topology—structure learning. Generally, exploration of this design space is a time consuming iterative process that requires close supervision by a human expert. Many studies provide guidelines for design choices such as network depth (Simonyan & Zisserman, 2014), layer width (Zagoruyko & Komodakis, 2016), building blocks (Szegedy et al., 2015), and connectivity (He et al., 2016; Huang et al., 2016). Based on these guidelines, these studies propose several meta-architectures, trained on huge volumes of data. These were applied to other tasks by leveraging the representational power of their convolutional layers and fine-tuning their deepest layers for the task at hand (Donahue et al., 2014; Hinton et al., 2015; Long et al., 2015; Chen et al., 2015; Liu et al., 2015). However, these meta-architectures may be unnecessarily large and require large computational power and memory for training and inference.

The problem of model structure learning has been widely researched for many years in the probabilistic graphical models domain. Specifically, Bayesian networks for density estimation and causal discovery (Pearl, 2009; Spirtes et al., 2000). Two main approaches were studied: score-based and constraint-based. Score-based approaches combine a scoring function, such as BDe (Cooper & Herskovits, 1992), with a strategy for searching in the space of structures, such as greedy equivalence search (Chickering, 2002). Adams et al. (2010) introduced an algorithm for sampling deep belief networks (generative model) and demonstrated its applicability to high-dimensional image datasets.

Constraint-based approaches (Pearl, 2009; Spirtes et al., 2000) find the optimal structures in the large sample limit by testing conditional independence (CI) between pairs of variables. They are generally faster than score-based approaches (Yehezkel & Lerner, 2009) and have a well-defined stopping criterion (e.g., maximal order of conditional independence). However, these methods are sensitive to errors in the independence tests, especially in the case of high-order CI tests and small training sets.

Motivated by these methods, we propose a new interpretation for depth and inter-layer connectivity in deep neural networks. We derive a structure learning algorithm such that a hierarchy of independencies in the input distribution is encoded in a deep generative graph, where lower-order independencies are encoded in deeper layers. Thus, the number of layers is automatically determined, which is a desirable virtue in any architecture learning method. We then convert the generative graph into a discriminative graph, demonstrating the ability of the latter to mimic (preserve conditional dependencies) of the former. In the resulting structure, a neuron in a layer is allowed to connect to neurons in deeper layers skipping intermediate layers. This is similar to the shortcut connection (Raiko et al., 2012), while our method derives it automatically. Moreover, neurons in deeper layers represent low-order (small condition sets) independencies and have a wide scope of the input, whereas neurons in the first layers represent higher-order (larger condition sets) independencies and have a narrower scope. An example of a learned structure, for MNIST, is given in Figure 1 ($X$ are image pixels).

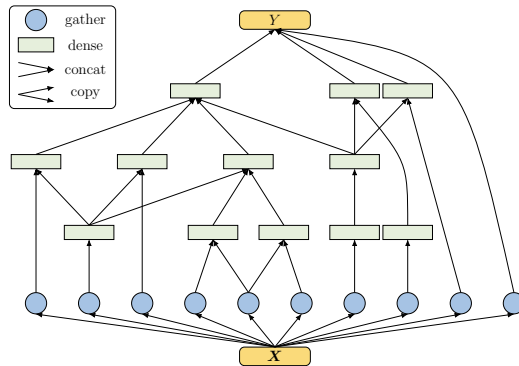

Figure 1: An example of a structure learned by our algorithm (classifying MNIST digits, 99.07% accuracy). Neurons in a layer may connect to neurons in any deeper layer. Depth is determined automatically. Each gather layer selects a subset of the input, where each input variable is gathered only once. A neural route, starting with a gather layer, passes through densely connected layers where it may split (copy) and merge (concatenate) with other routes in correspondence with the hierarchy of independencies identified by the algorithm. All routes merge into the final output layer.

The paper is organized as follows. We discuss related work in Section 2. In Section 3 we describe our method and prove its correctness in supplementary material Sec. A. We provide experimental results in Section 4, and conclude in Section 5.

## 2  Related Work

Recent studies have focused on automating the exploration of the design space, posing it as a hyper-parameter optimization problem and proposing various approaches to solve it. Miconi (2016) learns the topology of an RNN introducing structural parameters into the model and optimizing them along with the model weights by the common gradient descent methods. Smith et al. (2016) take a similar approach incorporating the structure learning into the parameter learning scheme, gradually growing the network up to a maximum size.

A common approach is to define the design space in a way that enables a feasible exploration process and design an effective method for exploring it. Zoph & Le (2016) (NAS) first define a set of hyper-parameters characterizing a layer (number of filters, kernel size, stride). Then they use a controller-RNN for finding the optimal sequence of layer configurations for a "trainee network". This is done using policy gradients (REINFORCE) for optimizing the objective function that is based on the accuracy achieved by the "trainee" on a validation set. Although this work demonstrates capabilities to solve large-scale problems (Imagenet), it comes with huge computational cost. In a following work, Zoph et al. (2017) address the same problem but apply a hierarchical approach. They use NAS to design network modules on a small-scale dataset (CIFAR-10) and transfer this knowledge to a large-scale problem by learning the optimal topology composed of these modules. Baker et al. (2016) use reinforcement learning as well and apply Q-learning with epsilon-greedy exploration strategy and experience replay. Negrinho & Gordon (2017) propose a language that allows a human expert to compactly represent a complex search-space over architectures and hyper-parameters as a tree and then use methods such as MCTS or SMBO to traverse this tree. Smithson et al. (2016) present a multi objective design space exploration, taking into account not only the classification accuracy but also the computational cost. In order to reduce the cost involved in evaluating the network's accuracy, they train a Response Surface Model that predicts the accuracy at a much lower cost, reducing the number of candidates that go through actual validation accuracy evaluation. Another common approach for architecture search is based on evolutionary strategies to define and search the design space. Real et al. (2017) and Miikkulainen et al. (2017) use evolutionary algorithm to evolve an initial model or blueprint based on its validation performance.

Common to all these recent studies is the fact that structure learning is done in a supervised manner, eventually learning a discriminative model. Moroeever, these approaches require huge compute resources, rendering the solution unfeasible for most applications given limited compute and time.

## 3   Proposed Method

**Preliminaries.** Consider $\boldsymbol{X} = \{X_i\}_{i=1}^N$ a set of observed (input) random variables, $\boldsymbol{H}$ a set of latent variables, and $Y$ a target (classification or regression) variable. Each variable is represented by a single node, and a single edge connects two distinct nodes. The parent set of a node $X$ in $G$ is denoted $\mathtt{Pa}(X;G)$, and the children set is denoted $\mathtt{Ch}(X;G)$. Consider four graphical models, $G$, $G_{\mathrm{inv}}$, $G_{\mathrm{dis}}$, and $g_X$. Graph $G$ is a generative DAG defined over $\boldsymbol{X} \cup \boldsymbol{H}$, where $\mathtt{Ch}(X;G) = \emptyset, \forall X \in \boldsymbol{X}$. Graph $G$ can be described as a layered deep Bayesian network where the parents of a node can be in any deeper layer and not restricted to the previous layer[1]. In a graph with $m$ latent layers, we index the deepest layer as $0$ and the layer connected to the input as $m-1$. The root nodes (parentless) are latent, $\boldsymbol{H}^{(0)}$, and the leaves (childless) are the observed nodes, $\boldsymbol{X}$, and $\mathtt{Pa}(\boldsymbol{X};G) \subset \boldsymbol{H}$. Graph $G_{\mathrm{inv}}$ is called a stochastic inverse of $G$, defined over $\boldsymbol{X} \cup \boldsymbol{H}$, where $\mathtt{Pa}(X;G_{\mathrm{inv}}) = \emptyset, \forall X \in \boldsymbol{X}$. Graph $G_{\mathrm{dis}}$ is a discriminative graph defined over $\boldsymbol{X} \cup \boldsymbol{H} \cup Y$, where $\mathtt{Pa}(X;G_{\mathrm{dis}}) = \emptyset, \forall X \in \boldsymbol{X}$ and $\mathtt{Ch}(Y;G_{\mathrm{dis}}) = \emptyset$. Graph $g_X$ is a CPDAG (a family of Markov equivalent Bayesian networks) defined over $\boldsymbol{X}$. Graph $g_X$ is generated and maintained as an internal state of the algorithm, serving as an auxiliary graph. The order of an independence relation between two variables is defined to be the condition set size. For example, if $X_1$ and $X_2$ are independent given $X_3$, $X_4$, and $X_5$ (d-separated in the faithful DAG $X_1 \perp\!\!\!\perp X_2 | \{X_3, X_4, X_5\}$), then the independence order is $|\{X_3, X_4, X_5\}| = 3$.

### 3.1   Key Idea

We cast the problem of learning the structure of a deep neural network as a problem of learning the structure of a deep (discriminative) probabilistic graphical model, $G_{\mathrm{dis}}$. That is, a graph of the form $\boldsymbol{X} \rightsquigarrow \boldsymbol{H}^{(m-1)} \rightsquigarrow \cdots \rightsquigarrow \boldsymbol{H}^{(0)} \rightarrow Y$, where "$\rightsquigarrow$" represent a sparse connectivity which we learn, and "$\rightarrow$" represents full connectivity. The joint probability factorizes as $P(\boldsymbol{X})P(\boldsymbol{H}|\boldsymbol{X})P(Y|\boldsymbol{H}^{(0)})$ and the posterior is $P(Y|\boldsymbol{X}) = \int P(\boldsymbol{H}|\boldsymbol{X})P(Y|\boldsymbol{H}^{(0)})d\boldsymbol{H}$, where $\boldsymbol{H} = \{\boldsymbol{H}^{(i)}\}_0^{m-1}$. We refer to the $P(\boldsymbol{H}|\boldsymbol{X})$ part of the equation as the *recognition network* of an unknown "true" underlying generative model, $P(\boldsymbol{X}|\boldsymbol{H})$. That is, the network corresponding to $P(\boldsymbol{H}|\boldsymbol{X})$ approximates the posterior (e.g., as in amortized inference). The key idea is to approximate the latents $\boldsymbol{H}$ that

generated the observed $\boldsymbol{X}$, and then use these values of $\boldsymbol{H}^{(0)}$ for classification. That is, avoid learning $G_{\text{dis}}$ directly and instead, learn a generative structure $\boldsymbol{X} \leftsquigarrow \boldsymbol{H}$, and then reverse the flow by constructing a stochastic inverse (Stuhlmüller et al., 2013) $\boldsymbol{X} \rightsquigarrow \boldsymbol{H}$. Finally, add $Y$ and modify the graph to preserve conditional dependencies ($G_{\text{dis}}$ can mimic $G$; $G_{\text{dis}}$ does not include sparsity that is not supported by $G$). Lastly, $G_{\text{dis}}$ is converted into a deep neural network by replacing each latent variable by a neural layer. We call this method B2N (Bayesian to Neural), as it learns the connectivity of a deep neural network through Bayesian network structure.

## 3.2 Constructing a Deep Generative Graph

The key idea of constructing $G$, the generative graph, is to recursively introduce a new latent layer, $\boldsymbol{H}^{(n)}$, after testing $n$-th order conditional independence in $\boldsymbol{X}$, and connect it, as a parent, to latent layers created by subsequent recursive calls that test conditional independence of order $n+1$. To better understand why deeper layer represent smaller condition independence sets, consider an ancestral sampling of the generative graph. First, the values of nodes in the deepest layer, corresponding to marginal independence, are sampled—each node is sampled *independently*. In the next layer, nodes can be sampled independently given the values of deeper nodes. This enables gradually factorizing ("disentangling") the joint distribution over $\boldsymbol{X}$. Hence, approximating the values of latents, $\boldsymbol{H}$, in the deepest layer provides us with statistically *independent* features of the data, which can be fed in to a single layer linear classifier. Yehezkel & Lerner (2009) introduced an efficient algorithm (RAI) for constructing a CPDAG over $\boldsymbol{X}$ by a recursive application of conditional independence tests with increasing condition set sizes. Our algorithm is based on this framework for testing independence in $\boldsymbol{X}$ and updating the auxiliary graph $g_X$.

Our proposed recursive algorithm for constructing $G$, is presented in Algorithm 1 (DeepGen) and a flow chart is shown in the supplementary material Sec. B. The algorithm starts with condition set $n = 0$, $g_X$ a complete graph (defined over $\boldsymbol{X}$), and a set of exogenous nodes, $\boldsymbol{X}_{\text{ex}} = \emptyset$. The set $\boldsymbol{X}_{\text{ex}}$ is exogenous to $g_X$ and consists of parents of $\boldsymbol{X}$. Note that there are two exit points, lines 4 and 14. Also, there are multiple recursive calls, lines 8 (within a loop) and 9, leading to multiple parallel recursive-traces, which will construct multiple generative flows rooted at some deeper layer.

The algorithm starts by testing the exit condition (line 2). It is satisfied if there are not enough nodes in $\boldsymbol{X}$ for a condition set of size $n$. In this case, the maximal depth is reached and an empty graph is returned (a layer composed of observed nodes). From this point, the recursive procedure will trace back, adding latent parent layers.

---

**Algorithm 1:** $G \longleftarrow \text{DeepGen}(g_X, \boldsymbol{X}, \boldsymbol{X}_{\text{ex}}, n)$

---

1 **DeepGen** $(g_X, \boldsymbol{X}, \boldsymbol{X}_{\text{ex}}, n)$

    **Input:** an initial CPDAG $g_X$ over endogeneous $\boldsymbol{X}$ & exogenous $\boldsymbol{X}_{\text{ex}}$ observed nodes, and a desired resolution $n$.

    **Output:** $G$, a latent structure over $\boldsymbol{X}$ and $\boldsymbol{H}$

2     **if** *the maximal indegree of $g_X(\boldsymbol{X})$ is below $n + 1$* **then**         ▷ exit condition

3         $G \longleftarrow$ an empty graph over $\boldsymbol{X}$         ▷ create a gather layer

4         **return** $G$

5     $g'_X \longleftarrow \text{IncSeparation}(g_X, n)$         ▷ $n$-th order independencies

6     $\{\boldsymbol{X}_{\text{D}}, \boldsymbol{X}_{\text{A}1}, \ldots, \boldsymbol{X}_{\text{A}k}\} \longleftarrow \text{SplitAutonomous}(\boldsymbol{X}, g'_X)$     ▷ identify autonomies

7     **for** $i \in \{1 \ldots k\}$ **do**

8         $G_{\text{A}i} \longleftarrow \text{DeepGen}(g'_X, \boldsymbol{X}_{\text{A}i}, \boldsymbol{X}_{\text{ex}}, n + 1)$         ▷ a recursive call

9     $G_{\text{D}} \longleftarrow \text{DeepGen}(g'_X, \boldsymbol{X}_{\text{D}}, \boldsymbol{X}_{\text{ex}} \cup \{\boldsymbol{X}_{\text{A}i}\}_{i=1}^k, n + 1)$     ▷ a recursive call

10     $G \longleftarrow (\bigcup_{i=1}^k G_{\text{A}i}) \cup G_{\text{D}}$         ▷ merge results

11     Create in $G$, $k$ latent nodes $\boldsymbol{H}^{(n)} = \{H_1^{(n)}, \ldots, H_k^{(n)}\}$     ▷ create a latent layer

12     Let $\boldsymbol{H}_{\text{A}i}^{(n+1)}$ and $\boldsymbol{H}_{\text{D}}^{(n+1)}$ be the sets of parentless nodes in $G_{\text{A}i}$ and $G_{\text{D}}$, respectively.

13     Set each $H_i^{(n)}$ to be a parent of $\{\boldsymbol{H}_{\text{A}i}^{(n+1)} \cup \boldsymbol{H}_{\text{D}}^{(n+1)}\}$     ▷ connect

14     **return** $G$

---

The procedure `IncSeparation` (line 5) disconnects (in $g_X$) conditionally independent variables in two steps. First, it tests dependency between $\boldsymbol{X}_{\text{ex}}$ and $\boldsymbol{X}$, i.e., $X \perp\!\!\!\perp X' | \boldsymbol{S}$ for every connected pair $X \in \boldsymbol{X}$ and $X' \in \boldsymbol{X}_{\text{ex}}$ given a condition set $\boldsymbol{S} \subset \{\boldsymbol{X}_{\text{ex}} \cup \boldsymbol{X}\}$ of size $n$. Next, it tests dependency within $\boldsymbol{X}$, i.e., $X_i \perp\!\!\!\perp X_j | \boldsymbol{S}$ for every connected pair $X_i, X_j \in \boldsymbol{X}$ given a condition set $\boldsymbol{S} \subset \{\boldsymbol{X}_{\text{ex}} \cup \boldsymbol{X}\}$ of size $n$. After removing the corresponding edges, the remaining edges are directed by applying two rules (Pearl, 2009; Spirtes et al., 2000). First, v-structures are identified and directed. Then, edges are continually directed, by avoiding the creation of new v-structures and directed cycles, until no more edges can be directed. Following the terminology of Yehezkel & Lerner (2009), we say that this function increases the graph d-separation resolution from $n-1$ to $n$.

The procedure `SplitAutonomous` (line 6) identifies autonomous sets, one descendant set, $\boldsymbol{X}_{\text{D}}$, and $k$ ancestor sets, $\boldsymbol{X}_{\text{A}1}, \ldots, \boldsymbol{X}_{\text{A}k}$ in two steps. First, the nodes having the lowest topological order are grouped into $\boldsymbol{X}_{\text{D}}$. Then, $\boldsymbol{X}_{\text{D}}$ is removed (temporarily) from $g_X$ revealing unconnected sub-structures. The number of unconnected sub-structures is denoted by $k$ and the nodes set of each sub-structure is denoted by $\boldsymbol{X}_{\text{A}i}$ ($i \in \{1 \ldots k\}$).

An autonomous set in $g_X$ includes all its nodes' parents (complying with the Markov property) and therefore a corresponding latent structure can be further learned independently, using a recursive call. Thus, the algorithm is called recursively and independently for the $k$ ancestor sets (line 8), and then for the descendant set, treating the ancestor sets as exogenous (line 9). This recursive decomposition of $\boldsymbol{X}$ is illustrated in Figure 2. Each recursive call returns a latent structure for each autonomous set. Recall that each latent structure encodes a generative distribution over the observed variables where layer $\boldsymbol{H}^{(n+1)}$, the last added layer (parentless nodes), is a representation of some input subset $\boldsymbol{X}' \subset \boldsymbol{X}$. Thus, latent variables, $\boldsymbol{H}^{(n)}$, are introduced as parents of the $\boldsymbol{H}^{(n+1)}$ layers (lines 11–13).

It is important to note that conditional independence is tested only between input variables, $\boldsymbol{X}$, and condition sets do not include latent variables. Conditioning on latent variables or testing independence between them is not required by our approach. A 2-layer toy-example is given in Figure 3.

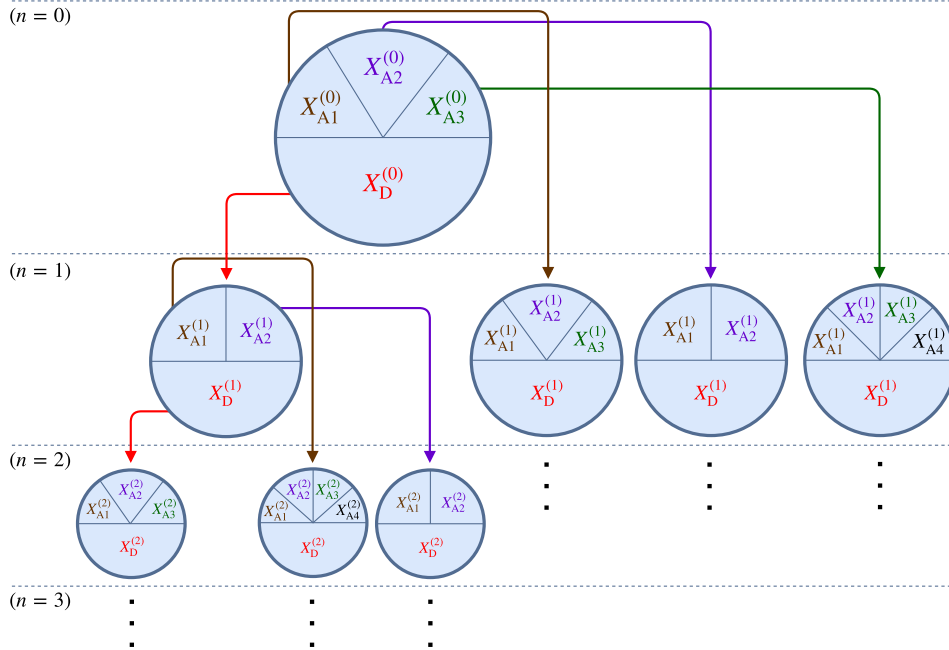

Figure 2: An example of a recursive decomposition of the observed set, $\boldsymbol{X}$. Each circle represents a distinct subset of observed variables (e.g., $\boldsymbol{X}_{\text{A}1}^{(1)}$ in different circles represents different subsets). At $n = 0$, a single circle represents all the variables. Each set of variables is split into autonomous ancestors $\boldsymbol{X}_{\text{A}i}^{(n)}$ and descendent $\boldsymbol{X}_{\text{D}}^{(n)}$ subsets. An arrow indicates a recursive call. Best viewed in color.

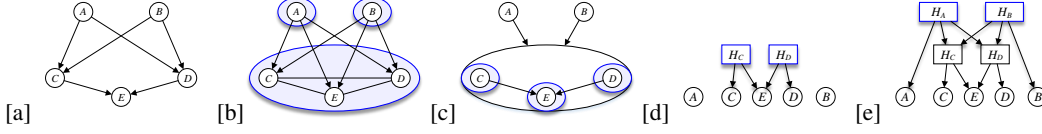

Figure 3: An example of learning a 2-layer generative model. [a] An example Bayesian network encoding the underlying independencies in $\boldsymbol{X}$. [b] $g_X$ after marginal independence testing ($n = 0$). Only $A$ and $B$ are marginally independent ($A \perp\!\!\!\perp B$). [c] $g_X$ after a recursive call to learn the structure of nodes $\{C, D, E\}$ with $n = 2$ ($C \perp\!\!\!\perp D|\{A, B\}$). Exit condition is met in subsequent recursive calls and thus latent variables are added to $G$ at $n = 2$ [d], and then at $n = 0$ [e] (the final structure).

### 3.3   Constructing a Discriminative Graph

We now describe how to convert $G$ into a discriminative graph, $G_{\mathrm{dis}}$, with target variable, $Y$ (classification/regression). First, we construct $G_{\mathrm{inv}}$, a graphical model that preserves all conditional dependencies in $G$ but has a different node ordering in which the observed variables, $\boldsymbol{X}$, have the highest topological order (parentless)—a stochastic inverse of $G$. Stuhlmüller et al. (2013) and Paige & Wood (2016) presented a heuristic algorithm for constructing such stochastic inverses. However, limiting $G_{\mathrm{inv}}$ to a DAG, although preserving all conditional dependencies, may omit many independencies and add new edges between layers. Instead, we allow it to be a projection of a latent structure (Pearl, 2009). That is, we assume the presence of *additional* hidden variables $\boldsymbol{Q}$ that are not in $G_{\mathrm{inv}}$ but induce dependency (for example, "interactive forks" (Pearl, 2009)) among $\boldsymbol{H}$. For clarity, we omit these variables from the graph and use bi-directional edges to represent the dependency induced by them. $G_{\mathrm{inv}}$ is constructed in two steps:

1. Invert the direction of all the edges in $G$ (invert inter-layer connectivity).
2. Connect each pair of latent variables, sharing a common child in $G$, with a bi-directional edge.

These steps ensure the preservation of conditional dependence.

**Proposition 1.** *Graph $G_{\mathrm{inv}}$ preserves all conditional dependencies in $G$ (i.e., $G \preceq G_{\mathrm{inv}}$).*

Note that conditional dependencies among $\boldsymbol{X}$ are not required to be preserved in $G_{\mathrm{inv}}$ and $G_{\mathrm{dis}}$ as these are observed variables (Paige & Wood, 2016).

Finally, a discriminative graph $G_{\mathrm{dis}}$ is constructed by replacing the bi-directional dependency relations in $G_{\mathrm{inv}}$ (induced by $\boldsymbol{Q}$) with explaining-away relations, which are provided by adding the observed class variable $Y$. Node $Y$ is set in $G_{\mathrm{dis}}$ to be the common child of the leaves in $G_{\mathrm{inv}}$ (latents introduced after testing marginal independencies in $\boldsymbol{X}$). See an example in Figure 4. This ensures the preservation of conditional dependency relations in $G_{\mathrm{inv}}$. That is, $G_{\mathrm{dis}}$, given $Y$, can mimic $G_{\mathrm{inv}}$.

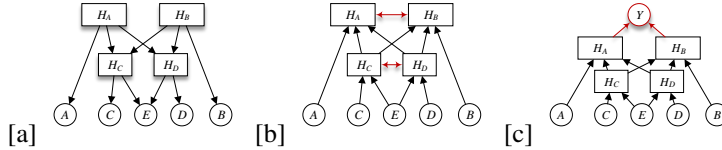

Figure 4: An example of the three graphs constructed by our algorithm: [a] a generative deep latent structure $G$, [b] its stochastic inverse $G_{\mathrm{inv}}$ (Stuhlmüller et al., 2013; Paige & Wood, 2016), and [c] a discriminative structure $G_{\mathrm{dis}}$ (target node $Y$ is added).

**Proposition 2.** *Graph $G_{\mathrm{dis}}$, conditioned on $Y$, preserves all conditional dependencies in $G_{\mathrm{inv}}$ (i.e., $G_{\mathrm{inv}} \preceq G_{\mathrm{dis}}|Y$).*

It follows that $G \preceq G_{\mathrm{inv}} \preceq G_{\mathrm{dis}}$ conditioned on $Y$.

**Proposition 3.** *Graph $G_{\mathrm{dis}}$, conditioned on $Y$, preserves all conditional dependencies in $G$ (i.e., $G \preceq G_{\mathrm{dis}}$).*

Details and proofs for all the propositions are provided in supplementary material Sec. A.

### 3.4 Constructing a Feed-Forward Neural Network

We construct a neural network based on the connectivity in $G_{\text{dis}}$. Sigmoid belief networks (Neal, 1992) have been shown to be powerful neural network density estimators (Larochelle & Murray, 2011; Germain et al., 2015). In these networks, conditional probabilities are defined as logistic regressors. Similarly, for $G_{\text{dis}}$ we may define for each latent variable $H' \in \boldsymbol{H}$, $p(H' = 1|\boldsymbol{X}') = \text{sigm}\left(\boldsymbol{W}'\boldsymbol{X}' + b'\right)$ where $\text{sigm}(x) = 1/(1 + \exp(-x))$, $\boldsymbol{X}' = \boldsymbol{Pa}(H'; G_{\text{dis}})$, and $(\boldsymbol{W}', b')$ are the parameters of the neural network. Nair & Hinton (2010) proposed replacing each binary stochastic node $H'$ by an infinite number of copies having the same weights but with decreasing bias offsets by one. They showed that this infinite set can be approximated by $\sum_{i=1}^{N} \text{sigm}(v-i+0.5) \approx \log(1+e^v)$, where $v = \boldsymbol{W}'\boldsymbol{X}' + b'$. They further approximate this function by $\max(0, v + \epsilon)$ where $\epsilon$ is a zero-centered Gaussian noise. Following these approximations, they provide an approximate probabilistic interpretation for the ReLU function, $\max(0, v)$. As demonstrated by Jarrett et al. (2009) and Nair & Hinton (2010), these units are able to learn better features for object classification in images.

In order to further increase the representational power, we represent each $H'$ by a set of neurons having ReLU activation functions. That is, each latent variable $H'$ in $G_{\text{dis}}$ is represented in the neural network by a fully-connected layer. Finally, the class node $Y$ is represented by a softmax layer.

## 4 Experiments

Our structure learning algorithm is implemented using BNT (Murphy, 2001) and runs efficiently on a standard desktop CPU (excluding neural network parameter learning). For the learned structures, all layers were allocated an equal number of neurons. Threshold for independence tests, and the number of neurons-per-layer were selected by using a validation set. In all the experiments, we used ReLU activations, ADAM (Kingma & Ba, 2015) optimization, batch normalization (Ioffe & Szegedy, 2015), and dropout (Srivastava et al., 2014) to all the dense layers. All optimization hyper-parameters that were tuned for the vanilla topologies were also used, without additional tuning, for the learned structures. In all the experiments, parameter learning was repeated five times where average and standard deviation of the classification accuracy were recorded. Only test-set accuracy is reported.

### 4.1 Learning the Structure of the Deepest Layers in Common Topologies

We evaluate the quality of our learned structures using five image classification benchmarks and seven common topologies (and simpler hand-crafted structures), which we call "vanilla topologies". The benchmarks and vanilla topologies are described in Table 1. Similarly to Li et al. (2017), we used the VGG-16 network that was previously modified and adapted for the CIFAR-10 dataset. This VGG-16 version contains significantly fewer parameters than the original one.

Table 1: Benchmarks and vanilla topologies used in our experiments. MNIST-Man and SVHN-Man topologies were manually created by us. MNIST-Man has two convolutional layer (32 and 64 filters each) and one dense layer with 128 neurons. SVHN-Man was created as a small network reference having reasonable accuracy (Acc.) compared to Maxout-NiN.

| Dataset | Id. | Topology | Description | Size | Acc. |
|---|---|---|---|---|---|
| | | | Vanilla Topology | | |
| MNIST (LeCun et al., 1998) | A | MNIST-Man | 32-64-FC:128 | 127K | 99.35 |
| SVHN (Netzer et al., 2011) | B | Maxout NiN | (Chang & Chen, 2015) | 1.6M | 98.10 |
| | C | SVHN-Man | 16-16-32-32-64-FC:256 | 105K | 97.10 |
| CIFAR 10 (Krizhevsky & Hinton, 2009) | D | VGG-16 | (Simonyan & Zisserman, 2014) | 15M | 92.32 |
| | E | WRN-40-4 | (Zagoruyko & Komodakis, 2016) | 9M | 95.09 |
| CIFAR 100 (Krizhevsky & Hinton, 2009) | F | VGG-16 | (Simonyan & Zisserman, 2014) | 15M | 68.86 |
| ImageNet (Deng et al., 2009) | G | AlexNet | (Krizhevsky et al., 2012) | 61M | 57.20 |

In preliminary experiments we found that, for SVHN and ImageNet, a small subset of the training data is sufficient for learning the structure. As a result, for SVHN only the basic training data is used (without the extra data), i.e., 13% of the available training data, and for ImageNet 5% of the training data is used. Parameters were optimized using all of the training data.

Convolutional layers are powerful feature extractors for images exploiting spatial smoothness properties, translational invariance and symmetry. We therefore evaluate our algorithm by using the first convolutional layers of the vanilla topologies as "feature extractors" (mostly below 50% of the vanilla network size) and then learning a deep structure, "learned head", from their output. That is, the deepest layers of the vanilla network, "vanilla head", is removed and replaced by a structure which is learned, in an unsupervised manner, by our algorithm[2]. This results in a new architecture which we train end-to-end. Finally, a softmax layer is added and the entire network parameters are optimized.

First, we evaluate the accuracy of the learned structure as a function of the number of parameters and compare it to a densely connected network (fully connected layers) having the same depth and size (Figure 5). For SVHN, we used the Batch Normalized Maxout Network-in-Network topology (Chang & Chen, 2015) and removed the deepest layers starting from the output of the second NiN block (MMLP-2-2). For CIFAR-10, we used the VGG-16 and removed the deepest layers starting from the output of conv.7 layer. It is evident that accuracies of the learned structures are significantly higher (error bars represent 2 standard deviations) than those produced by a set of fully connected layers, especially in cases where the network is limited to a small number of parameters.

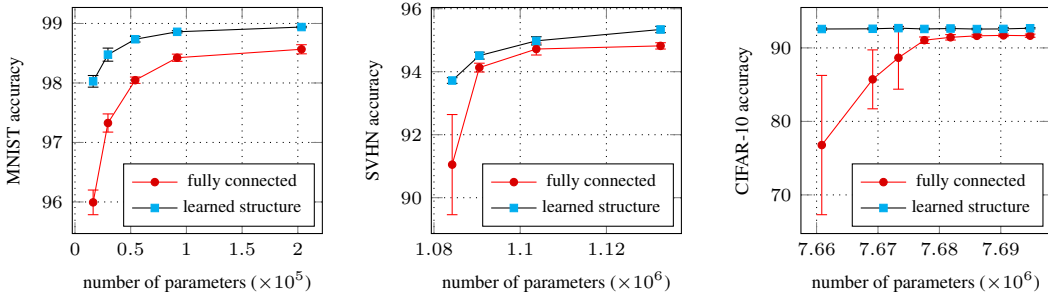

Figure 5: Classification accuracy of MNIST, SVHN, and CIFAR-10, as a function of network size. Error bars indicate two standard deviations.

Next, in Figure 6 and Table 2 we provide a summary of network sizes and classification accuracies, achieved by replacing the deepest layers of common topologies (vanilla) with a learned structure. In all the cases, the size of the learned structure is significantly smaller than that of the vanilla topology.

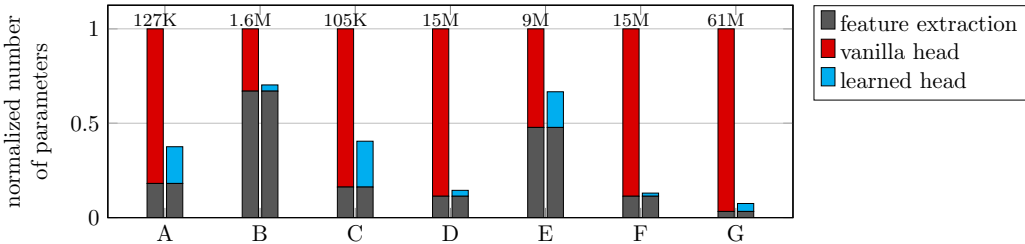

Figure 6: A comparison between the vanilla and our learned structure (B2N), in terms of normalized number of parameters. The first few layers of the vanilla topology are used for feature extraction. Stacked bars refer to either the vanilla or our learned structure. The total number of parameters of the vanilla network is indicated on top of each stacked bar.

## 4.2 Comparison to Other Methods

Our structure learning algorithm runs efficiently on a standard desktop CPU, while providing structures with competitive classification accuracies and network sizes. First, we compare our method to the NAS algorithm (Zoph & Le, 2016). NAS achieves for CIFAR-10 an error rate of 5.5% with a network of size 4.2M. Our method, using the feature extraction of the WRN-40-4 network, achieves this same error rate with a 26% smaller network (3.1M total size). Using the same feature extraction, the lowest classification error rate achieved by our algorithm for CIFAR 10 is 4.58% with a network of size 6M whereas the NAS algorithm achieves an error rate of 4.47% with a network of size 7.1M. Recall that the NAS algorithm requires training thousands of networks using hundreds of GPUs, which is impractical for most real-world applications.

When compared to recent pruning methods, which focus on reducing the number of parameters in a pre-trained network, our method demonstrates state-of-the-art reduction in parameters. Recently reported results are summarized in Table 3. It is important to note that although these methods prune all the network layers, whereas our method only replaces the network head, our method was found significantly superior. Moreover, pruning can be applied to the feature extraction part of the network which may further improve parameter reduction.

Table 2: Parameter reduction ratio (vanilla size/learned size) and difference in classification accuracy (Acc. Diff.=learned−vanilla, higher is better). "Full"=feature extration+head.

| Id. | Acc. Diff. | Param. Reduc. | |
|---|---|---|---|
| | | Full | Head |
| A | $+0.10 \pm 0.04$ | $2.7\times$ | $4.2\times$ |
| B | $-0.40 \pm 0.05$ | $1.4\times$ | $10.0\times$ |
| C | $-0.86 \pm 0.05$ | $2.5\times$ | $3.5\times$ |
| D | $+0.29 \pm 0.14$ | $7.0\times$ | $28.3\times$ |
| E | $+0.33 \pm 0.14$ | $1.5\times$ | $2.8\times$ |
| F | $+0.05 \pm 0.17$ | $7.7\times$ | $53.2\times$ |
| G | $+0.00 \pm 0.03$ | $13.3\times$ | $23.0\times$ |

Table 3: Parameter reduction ratio (vanilla/learned size) compared to recent pruning methods (reducing the size of a pre-trained network with *minimal* accuracy degradation). Results indicated by "acc. deg." correspond to accuracy degradation after pruning.

| Network | Method | Reduction |
|---|---|---|
| VGG-16 (CIFAR-10) | Li et al. (2017) | $3\times$ |
| | Ayinde & Zurada (2018) | $4.6\times$ |
| | Ding et al. (2018) | *(acc. deg.)* $5.4\times$ |
| | Huang et al. (2018) | *(acc. deg.)* $6\times$ |
| | B2N (our) | $\mathbf{7\times}$ |
| AlexNet (ImageNet) | Denton et al. (2014) | $5\times$ |
| | Yang et al. (2015) | $3.2\times$ |
| | Han et al. (2015, 2016) | $9\times$ |
| | Manessi et al. (2017) | *(acc. deg.)* $12\times$ |
| | B2N (our) | $\mathbf{13.3\times}$ |

## 5 Conclusions

We presented a principled approach for learning the structure of deep neural networks. Our proposed algorithm learns in an unsupervised manner and requires small computational cost. The resulting structures encode a hierarchy of independencies in the input distribution, where a node in one layer may connect to another node in any deeper layer, and network depth is determined automatically.

We demonstrated that our algorithm learns small structures, and maintains classification accuracies for common image classification benchmarks. It is also demonstrated that while convolution layers are very useful at exploiting domain knowledge, such as spatial smoothness, translational invariance, and symmetry, in some cases, they are outperformed by a learned structure for the deeper layers. Moreover, while the use of common topologies (meta-architectures), for a variety of classification tasks is computationally inefficient, we would expect our approach to learn smaller and more accurate networks for each classification task, uniquely.

As only unlabeled data is required for learning the structure, we expect our approach to be practical for many domains, beyond image classification, such as knowledge discovery, and plan to explore the interpretability of the learned structures. Casting the problem of learning the connectivity of deep neural network as a Bayesian network structure learning problem, enables the development of new principled and efficient approaches. This can lead to the development of new topologies and connectivity models, and can provide a greater understanding of the domain. One possible extension to our work which we plan to explore, is learning the connectivity between feature maps in convolutional layers.

## Footnotes

[1]This differs from the common definition of deep belief networks (Hinton et al., 2006; Adams et al., 2010) where the parents are restricted to the next layer.

[2]We also learned a structure for classifying MNIST digits directly from image pixels, without using convolutional layers for feature extraction. The resulting network structure (Figure 1), achieves an accuracy of 99.07%, whereas a network with 3 fully-connected layers achieves 98.75%.

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
