[Supplementary Material]

# Supplementary: Constructing Deep Neural Networks by Bayesian Network Structure Learning

**Raanan Y. Rohekar**
Intel AI Lab
raanan.yehezkel@intel.com

**Shami Nisimov**
Intel AI Lab
shami.nisimov@intel.com

**Yaniv Gurwicz**
Intel AI Lab
yaniv.gurwicz@intel.com

**Guy Koren**
Intel AI Lab
guy.koren@intel.com

**Gal Novik**
Intel AI Lab
gal.novik@intel.com

## A  Preservation of Conditional Dependence

We prove that conditional dependence relations encoded by the generative structure $G$ are preserved by the discriminative structure $G_{\mathrm{dis}}$ conditioned on the class $Y$. That is, $G_{\mathrm{dis}}$ conditioned on $Y$ can mimic $G$; denoted by $G \preceq G_{\mathrm{dis}}|Y$, a preference relation. While the parameters of a model can learn to mimic conditional independence relations that are not expressed by the graph structure, they are not able to learn conditional dependence relations (Pearl, 2009).

**Proposition 1.** *Graph $G_{\mathrm{inv}}$ preserves all conditional dependencies in $G$ (i.e., $G \preceq G_{\mathrm{inv}}$).*

*Proof.* Graph $G_{\mathrm{inv}}$ can be constructed using the procedures described by Stuhlmüller et al. (2013) where nodes are added, one-by-one, to $G_{\mathrm{inv}}$ in a reverse topological order (lowest first) and connected (as a child) to existing nodes in $G_{\mathrm{inv}}$ that d-separate it, according to $G$, from the remainder of $G_{\mathrm{inv}}$. Paige & Wood (2016) showed that this method ensures $G \preceq G_{\mathrm{inv}}$, the preservation of conditional dependence. We set an equal topological order to every pair of latents $(H_i, H_j)$ sharing a common child in $G$. Hence, jointly adding nodes $H_i$ and $H_j$ to $G_{\mathrm{inv}}$, connected by a bi-directional edge, requires connecting them (as children) only to their children and the parents of their children ($H_i$ and $H_j$ themselves, by definition) in $G$. That is, without loss of generality, node $H_i$ is d-separated from the remainder of $G_{\mathrm{inv}}$ given its children in $G$ and $H_j$. ∎

It is interesting to note that the stochastic inverse $G_{\mathrm{inv}}$, constructed without adding inter-layer connections, preserves all conditional dependencies in $G$.

**Proposition 2.** *Graph $G_{\mathrm{dis}}$, conditioned on $Y$, preserves all conditional dependencies in $G_{\mathrm{inv}}$ (i.e., $G_{\mathrm{inv}} \preceq G_{\mathrm{dis}}|Y$).*

*Proof.* It is only required to prove that the dependency relations that are represented by bi-directional edges in $G_{\mathrm{inv}}$ are preserved in $G_{\mathrm{dis}}$. The proof follows directly from the d-separation criterion (Pearl, 2009). A latent pair $\{H, H'\} \subset \boldsymbol{H}^{(n+1)}$, connected by a bi-directional edge in $G_{\mathrm{inv}}$, cannot be d-separated by any set containing $Y$, as $Y$ is a descendant of a common child of $H$ and $H'$. In Algorithm 1-line 16, a latent in $\boldsymbol{H}^{(n)}$ is connected, as a child (as a parent in $G$), to latents $\boldsymbol{H}^{(n+1)}$, and $Y$ to $\boldsymbol{H}^{(0)}$. ∎

We formulate $G_{\mathrm{inv}}$ as a projection of another latent model (Pearl, 2009) where bi-directional edges represent dependency relations induced by latent variables $\boldsymbol{Q}$. We construct a discriminative model by considering the effect of $\boldsymbol{Q}$ as an explaining-away relation induced by the target node $Y$. Thus, conditioned on $Y$, the discriminative graph $G_{\mathrm{dis}}$ preserves all conditional (and marginal) dependencies in $G_{\mathrm{inv}}$.

**Proposition 3.** *Graph $G_{\text{dis}}$, conditioned on $Y$, preserves all conditional dependencies in $G$ (i.e., $G \preceq G_{\text{dis}}$).*

*Proof.* It immediately follows from Propositions 1 & 2 that $G \preceq G_{\text{inv}} \preceq G_{\text{dis}}$ conditioned on $Y$. ∎

Thus $G \preceq G_{\text{inv}} \preceq G_{\text{dis}}$ conditioned on $Y$.

# B  Flowchart

Figure 1: Flowchart of the DeepGen algorithm.