[Reviews · NeurIPS 2018]

Reviewer 1



The presented method learns a structure of a deep ANN by first learning a BN and then constructing the ANN from this BN. The authors state that they "propose a new interpretation for depth and inter-layer connectivity in deep neural networks". Neurons in deep layers represent low-order conditional independencies (ie small conditioning set) and those in 'early' (non-deep) layers represent high-order CI relationships. These are all CI relations in the "X" ie the input vector of (observed) random variables. Perhaps I am missing something here but I could not find an argument as to why this is a principled way to build deep ANNs with good performance. After all, when we actually use an ANN to make a prediction, all the X input variables will be observed, so it is not obvious to me why we should care about CI relations between the components of X. So we are using unsupervised learning to help supervised learning. Nothing wrong with that, but I think we need more about how the former helps the latter in this case. The actual process of geneating the ANN structure is clearly described. Since the ultimate goal of the BN learning is somewhat unusual so is the BN learning algorithm - essentially we add latent variables to represent the (estimated) CI relations in the input. Although I can't see why this is a good method for making ANN structure, perhaps it just is, so the empirical evaluation is very important. 5 image classification and 7 other benchmarks are used. Not too surprisingly the empirical results are positive: the learned (more sparse) structures do better than fully connected ones, particularly with smaller training sets. The authors also (in Section 4.2) compare to other methods: getting comparable accuracies to NAS but with smaller networks. I think this is promising work and reasonable results have been produced but I judge that for publication in NIPS we need a fuller account of why/when this method should work. SPECIFIC POINTS Using a threshold for the CI tests set by a validation set is a good approach (certainly better than many constraint-based approaches where some arbitrary threshold is set and then CI test results on finite data are treated as 100% reliable). However, the authors do not address the issue of the inevitable errors in CI test results, despite flagging up this issue on lines 45-46. line 255: smaller number of -> fewer AFTER READING AUTHOR FEEDBACK The authors have provided a better explanation of the rationale of their approach, which I hope will be included in any future version, and I have upped my rating appropriately.

Reviewer 2



This work presents an approach to learning deep neural network architectures by drawing on ideas from Bayesian network structure learning. Layers of the network are established using nth order conditional independencies, and connections may appear between non-adjacent layers. This is an interesting idea that may, in addition to the efficiency arguments made by the authors, have benefits in terms of interpretability and identifiability of a network. The empirical results are satisfying in terms of the overall reduction in the number of parameters while maintaining predictive performance. The work is mostly well-presented but there are some details lacking; e.g. I cannot find the details of the independence test that is used (perhaps this is standard for those in the area, but I am not so familiar with it). I am unable to comment with great confidence on the novelty of the work. The authors do discuss related work, which suggests that at least part of the novelty of this approach is that it is unsupervised, while existing approaches are supervised. I am not familiar enough with the literature in this specific area to verify that claim. It does appear that it would quite cheap computationally, however.

Reviewer 3



The paper presents neural structure learning as a Bayesian structure learning approach. It first learns a DAG generative graph over the input distribution with latent variables in an unsupervised manner. Then it computes a stochastic inverse of the learned graph and finally converts it into a discriminative graph(adds label). Each latent variable in the learned structure is replaced by an MLP. The learned model is trained for the task. Several Concerns: 1. It is an interesting approach to model structure as a Bayesian network, however the advantages over baselines are not unclear since the experiments only show the structure learning of last layers. This makes the method rather a parameter reduction technique than structure learning. Why can't the experiments show an entire structure learned from scratch? 2. The current experiments use features extracted from pre-trained networks, not completely unsupervised. 3. The scalability in comparison to Neural Architecture Search (NAS) claim isn't supported by experiments; not accounting for the fact that NAS learns entire structure in comparison to only the deepest layers from a trained network in the experiments. 4. Can only be applied to the deepest layers for y label?